# Constructing of predictive model for the surgical effect of patients with cleft lip and palate

Na Liu[1,2], Jingyuan Yang[1]*, Fang Tan[3], Haijian Zhu[2]

**1** Department of Epidemiology and Health Statistics, School of Public Health and Health, Guizhou Medical University, Guiyang, Guizhou Province, China, **2** Department of Administrative Management, Guiyang Stomatological Hospital, Guiyang, Guizhou Province, China, **3** Department of Oral and Maxillofacial Surgery, Guiyang Stomatological Hospital, Guiyang, Guizhou Province, China

* yangjingyuan@gmc.edu.cn

## Abstract

### Objective

To explore effective factors of surgical effect for patients with cleft lip and palate, and to construct the predictive model of surgical effect, which provide reference for improving the effect of cleft lip and palate surgery.

### Methods

This study has been ethically reviewed and approved by the Medical Ethics Committee of Guiyang Stomatological Hospital before the study began.A total of 997 cases of cleft lip and palate surgical treatment in Guiyang Stomatological Hospital from 2015 to 2020 were collected. Logistic regression analysis was used to analyze the factors influencing the surgical outcome, and a score system was established by assigning values to the influencing factors using the nomogram. Data of 110 patients were verified, and decision curve analysis was used to evaluate the predicted results.

### Results

Logistic regression analysis showed that the number of surgeries, surgical methods, breast milk, prenatal examination, nutrition during pregnancy and labor intensity during pregnancy were independent risk factors for poor surgical results (all P<0.05). The predictive model was built by including the number of surgeries, surgical methods, breast milk, prenatal examination, nutrition and labor intensity during pregnancy into the predictive scoring system. The critical value was 273, the area under ROC curve (AUC) was 0.733(95% CI:0.704~0.76), the sensitivity was 89.57%, and the specificity was 48.14%.When the external validation data of 110 patients were brought into the score, the AUC of poor diagnostic value reached 74.5%, P<0.05, which was close to the modeling accuracy of 73.3%.

### Conclusion

This study constructed a predictive model of surgical effect for patients with cleft lip and palate, which can be used for the clinical prediction of cleft lip and palate patients in Guizhou Province.

**Data Availability Statement:** All relevant data are within the paper and its Supporting Information files.

**Funding:** The authors received no specific funding for this work.

**Competing interests:** The authors have declared that no competing interests exist.

## Introduction

Cleft lip and palate is one of the most common congenital malformations of the oral and maxillofacial region, which is caused by the failure to achieve normal facial fusion due to pathogenic factors that affect the facial fusion during its embryonic development [1], and different degrees of malformation can cause different degrees of functional impairment, such as chewing, sucking, swallowing, speech, expression and appearance defects, which seriously affect the quality of life of the affected children. Cleft lip and palate are divided into cleft lip and palate, cleft lip, and cleft palate according to the site of the cleft [2]. China is a country with a high prevalence of cleft lip and palate, with an overall prevalence of about 1.67‰ [3], and Guizhou Province is one of the regions with a high prevalence of cleft lip and palate, with an incidence of about 1.92‰ [4]. Currently, treatment is mainly performed through surgical procedures, but there are still more problems of secondary deformities after surgery, such as obvious scarring and nasal collapse [5]. Regarding surgical outcomes, the main focus is on the observation of clinical postoperative outcomes, while the prediction of preoperative outcomes lacks corresponding tools, and the establishment of surgery-related predictive indexes would provide an important reference for patients' families and surgeons. To this end, this study collected data related to the outcome of patients who had completed cleft lip and palate surgery in Guiyang Stomatological Hospital. It analyzed the factors influencing the surgical outcome and established a predictive assessment method for surgical outcome using column line diagrams, and constructed a predictive index system to play a role in improving the outcome of cleft lip and palate surgery, to improve the satisfaction of cleft lip and palate patients and their families with the surgical outcome and promoting the communication and cooperation between doctors and patients.

## Materials and methods

### Study subjects

Cases of cleft lip and palate treated surgically by oral and maxillofacial surgery in an oral hospital in Guiyang.The purpose of information collection should be explained in detail to all subjects before collection, and oral informed consent should be obtained from all subjects.

The study was examined and approved by the Ethics Committee of Guiyang Stomatological. Hospital.Ethics approval number:GYSKLL-KY-20220107-01.

Inclusion criteria: patients with non-syndromic cleft lip and palate; no other acute or chronic somatic diseases; no congenital malformations with other systems.

Exclusion criteria: patients with incomplete information.

### Study methods

**Data collection.** The medical records of patients with cleft lip and palate surgery in a Stomatological Hospital in Guiyang City between 2015 and 2020 were retrospectively collected, and information collection forms were established, including gender, age, ethnicity, birth weight, history of cleft lip, health status, medical history status, fetal status, mode of delivery, breastfeeding status, obstetric examination status, pregnancy status, duration of surgery, number of operations, and mode of surgery, etc.

**Evaluation criteria of surgical effect.** The postoperative outcome evaluation used now mainly referred to the practice of West China College of Stomatology at Sichuan University was referred to the Asher-McDade scale [6] and Mortier PB scale [7], respectively. The Asher-McDade scale mainly included the evaluation of nasal morphology, nasal symmetry, upper lip contour, lip peak, etc. The evaluation of the Mortier PB scale mainly includes the evaluation of excessive mucosal tissue, insufficient mucosal height, incision marks, thin red lip on the

affected side, narrow human middle, too long white lip, too short white lip, muscle break, obvious scar, etc.

**Structure of the surgical outcome evaluation team and tracking time.** The evaluation team was mainly composed of a physician with a senior title, a physician with an intermediate or junior title, and a nursing staff, and the tracking lasted for one year through outpatient follow-up, remote video, or photo follow-up. Measurements were taken through preoperative and postoperative photos to determine whether the symmetry and aesthetics of the nasolabial shape were restored after cleft lip surgery; the surgical status of cleft palate patients is judged by the degree of velopharyngeal closure, the function of an oropharyngeal area and the verbal ability. At the same time, the satisfaction of patients and their families with the surgery is also an evaluation indicator. The evaluation team scored based on the above indicators and finally made a comprehensive determination of the evaluation results of the three. 1 is very good, 2 is good, 3 is common, 4 is bad, and 5 is very bad.

**Development of predictive scoring tools.** The influencing factors of the surgical effect were analyzed according to the case-control study. The variables that were significant in the univariate analysis were included in the multivariate analysis. The results of the multivariate analysis were scored by using the column chart, and the prediction model was built. Verification truncation values were established to evaluate the predicted outcomes compared with actual surgical outcomes. And validate with external data.

**Statistical methods.** All data were statistically analyzed using SPSS17.0 statistical software and R software, and the chi-square test was used to compare the difference in the rate of poor surgical outcome in different subgroups, and logistic regression analysis was applied to analyze the influence of relevant factors on surgical outcome, and Fisher's exact test was used for one-way analysis with sample size $n < 40$ or theoretical frequency $T < 1$, to compare good and poor surgical The difference of relevant factors in patients, the prediction score system was constructed by using nomogram column line plot, the value of its prediction was evaluated by using ROC curve, the sensitivity and specificity of prediction effect of prediction score were calculated, and the prediction results were evaluated by using decision curve analysis (DCA) and clinical impact curve analysis (DCA), and the test level $\alpha$ was taken as 0.05.

## Results

### General information about the subjects

Among the 997 cleft lip and palate patients collected, 613 were males and 384 were females, with an age range of 2 to 40 years. 318 cases (32%) were cleft lip alone (CL); 169 cases (17%) were cleft palate alone (CP); 510 cases (51%) were cleft lip combined with cleft palate (CL+P), of which 882 cases (88.5%) were evaluated as good after surgery; 115 cases (11.5%) were evaluated as poor. 115 cases, accounting for 11.5% (as Tables 1 and 2).

### Analysis of factors influencing surgical outcomes

**Single-factor analysis.** Single-factor analysis was used to compare the differences in factors associated with good and poor surgical patients (as Table 3), and the results showed that

**Table 1. Type and composition ratio of cleft lip and palate.**

| Type | Number of cases | Composition ratio (%) |
|------|-----------------|------------------------|
| Cleft lip alone(CL) | 318 | 32 |
| Cleft lip combined with cleft palate(CL+P) | 510 | 51 |
| Cleft palate alone(CP) | 169 | 17 |
| Total | 997 | 100 |

**Table 2. Composition ratio of cleft lip and palate surgical outcomes.**

| Postoperative outcome | Number of cases | Composition ratio (%) |
|---|---|---|
| Good | 882 | 88.5 |
| Unsatisfactory | 115 | 11.5 |
| Total | 997 | 100 |

there were statistical differences (P<0.05) between poor and good surgeries in terms of the number of surgeries, surgical methods, breast milk, obstetric examinations, pregnancy nutrition, and labor intensity during pregnancy, so these factors needed to be screened out for subsequent multifactor analysis and the establishment of an index system.

**Multi-factor analysis.** All the variables screened out in the univariate analysis of the rate of unsatisfactory surgical outcome, P<0.2 or 0.1 continued to select binary logistic regression for multifactor analysis, and the results showed (as Table 4) that the number of surgeries could significantly affect the surgical outcome, P<0.05, OR was 0.563, indicating that the more the number of surgeries, the lower the probability of unsatisfactory surgical outcomes, the surgical method could significantly affect the surgical The OR was 16.597 and 15.649, respectively, meaning that the probability of unsatisfactory surgical outcomes were 16.597 and 15.649 times higher for the modified Lan method and the lower triangular flap method than for the other surgical methods; breast milk could significantly affect the surgical outcome, P<0.05, indicating that The probability of having a unsatisfactory surgery was significantly higher for breast-fed than for non-breast-fed, with an OR of 1.935, implying that the rate of unsatisfactory surgical outcome for breast milk was 1.935 times higher than that for non-breast milk; maternal examination could significantly affect the surgical outcome, with P<0.05 and an OR of 2.619, implying that the rate of unsatisfactory surgical outcome was 2.619 times higher for non-maternal examination; pregnancy nutrition could significantly affect the surgical outcome, with P<0.05 and an OR 1.609 and 3.56, respectively, implying that those with moderate or poor nutrition during pregnancy were 1.609 and 3.56 times more likely to have poor surgical outcomes than those with good nutrition during pregnancy; labor intensity during pregnancy significantly affected surgical outcomes, P<0.05, OR 1.851 and 2.152, respectively, implying that those with moderate or heavy labor intensity during pregnancy were 1.851 and 2.512 times more likely to have poor surgical outcomes than those with light labor intensity during pregnancy. 1.851, 2.512 times.

**Predictive scoring of surgical outcomes.** After exploring the influencing factors of surgical outcome, the nomogram statistical method was used to analyze and establish the predictive scoring system of surgical outcome, and the results were as follows (as Fig 1,Tables 5 and 6).

**Diagnostic surgical outcome ROC curve.** According to the probability of poor surgical outcome corresponding to the scoring system, a cut point of 50%, i.e, a patient's score greater than 273, means that the patient will have a poor surgical outcome. Verifying the prediction accuracy of the scoring system with c-index = 73.36%, the scores were brought into the patient data to obtain the diagnostic ROC curve of the nomo score on the surgical outcome as follows. (as Fig 2).

**Calibration plot of the scoring system.** To verify the accuracy of the above scoring system, the following calibration plot (calibration plot) is drawn as follows. (as Fig 3).

The intercept = 0.2846, slope = 0.8357, and it can be seen from the following plot that the prediction results and the diagonal line fit together, which means the prediction results are more accurate.

**Decision curve analysis (DCA).** The predictive scoring system established by nomogram contains six indicators, namely, the number of surgeries, surgical method, breast milk, delivery

**Table 3. Single-factor analysis affecting surgical outcome.**

| Category | Good | Poor | T/c2 | P |
|---|---|---|---|---|
| Duration of surgery | 72.33±29.29 | 75.77±32.34 | -1.17 | 0.242 |
| Number of surgeries | 1.39±0.68 | 1.29±0.49 | 2.026 | 0.044 |
| Birth weight | 3.2±0.56 | 3.16±0.52 | 0.795 | 0.427 |
| Surgical method | | | | |
| Modified Lang method | 325(85.3%) | 56(14.7%) | 26.637 | <0.001 |
| Inferior triangular flap method | 229(83.6%) | 45(16.4%) | | |
| Rotational propulsion method | 239(94.8%) | 13(5.2%) | | |
| Other | 69(98.6%) | 1(1.4%) | | |
| History of cleft lip | | | | |
| None | 787(88.7%) | 100(11.3%) | 2.288 | 0.13 |
| Yes | 75(83.3%) | 15(16.7%) | | |
| Health status | | | | |
| Good | 855(88.1%) | 115(11.9%) | - | 1△ |
| Adverse | 7(100%) | 0(0%) | | |
| Blood transfusion history | | | | |
| None | 856(88.2%) | 114(11.8%) | - | 0.585△ |
| Yes | 6(85.7%) | 1(14.3%) | | |
| History of infectious disease | | | | |
| None | 860(88.4%) | 113(11.6%) | - | 0.07△ |
| Yes | 2(50%) | 2(50%) | | |
| History of vaccination | | | | |
| None | 15(100%) | 0(0%) | - | 0.24△ |
| Yes | 847(88%) | 115(12%) | | |
| History of allergy | | | | |
| None | 836(88.3%) | 111(11.7%) | - | 0.772△ |
| Yes | 26(86.7%) | 4(13.3%) | | |
| Clinical manifestations | | | | |
| None | 837(88.3%) | 111(11.7%) | - | 0.767△ |
| Yes | 25(86.2%) | 4(13.8%) | | |
| History of trauma | | | | |
| None | 848(88.4%) | 111(11.6%) | - | 0.152△ |
| Yes | 14(77.8%) | 4(22.2%) | | |
| Other surgical histories | | | | |
| None | 840(88.2%) | 112(11.8%) | - | 1△ |
| Yes | 22(88%) | 3(12%) | | |
| Number of births | | | | |
| First birth | 335(88.4%) | 44(11.6%) | 1.921 | 0.383 |
| Second child | 362(89.4%) | 43(10.6%) | | |
| Third child and above | 165(85.5%) | 28(14.5%) | | |
| Full-term or not | | | | |
| Full term | 720(88.8%) | 91(11.2%) | 1.39 | 0.238 |
| Not full term | 142(85.5%) | 24(14.5%) | | |
| Delivery | | | | |
| Normal birth | 158(98.1%) | 3(1.9%) | - | 1△ |
| Cesarean section | 11(100%) | 0(0%) | | |
| Breastfeeding | | | | |

(*Continued*)

**Table 3.** (Continued)

| Category | Good | Poor | T/c2 | P |
|---|---|---|---|---|
| No | 504(90.5%) | 53(9.5%) | 6.347 | 0.012 |
| Yes | 358(85.2%) | 62(14.8%) | | |
| Maternal examination | | | | |
| Done | 838(88.7%) | 107(11.3%) | - | 0.044△ |
| Not done | 24(75%) | 8(25%) | | |
| Nutrition during pregnancy | | | | |
| Good | 740(89.2%) | 90(10.8%) | 8.652 | 0.013 |
| Medium | 105(85.4%) | 18(14.6%) | | |
| Poor | 17(70.8%) | 7(29.2%) | | |
| Spirituality during pregnancy | | | | |
| Good | 19(79.2%) | 5(20.8%) | 1.999 | 0.368 |
| Medium | 643(88.3%) | 85(11.7%) | | |
| Poor | 200(88.9%) | 25(11.1%) | | |
| Labor intensity during pregnancy | | | | |
| Light | 629(90.6%) | 65(9.4%) | 13.775 | 0.001 |
| Medium | 89(84%) | 17(16%) | | |
| Heavy | 144(81.4%) | 33(18.6%) | | |
| Trauma during pregnancy | | | | |
| None | 857(88.3%) | 114(11.7%) | - | 0.529△ |
| Yes | 5(83.3%) | 1(16.7%) | | |
| Pregnancy illness | | | | |
| None | 802(88.3%) | 106(11.7%) | 0.116 | 0.734 |
| Yes | 60(87%) | 9(13%) | | |
| Pregnancy vomiting | | | | |
| Light | 310(86.1%) | 50(13.9%) | 3.171 | 0.205 |
| Medium | 334(88.6%) | 43(11.4%) | | |
| Heavy | 218(90.8%) | 22(9.2%) | | |
| Parental drinking | | | | |
| None | 28(100%) | 0(0%) | - | 0.586△ |
| Yes | 92(94.8%) | 5(5.2%) | | |
| Folic acid | | | | |
| Not Taken | 55(93.2%) | 4(6.8%) | - | 0.188△ |
| Have taken | 65(98.5%) | 1(1.5%) | | |

Note: △ indicates Fisher's exact test; T/c2, indicates the statistic of T-test, and c2 indicates the statistic of chi-square test.

examination, nutrition during pregnancy, and labor intensity during pregnancy, i.e, the predictive scoring system established by nomogram is based on the joint prediction of the six indicators, and on this basis, the net patient benefit (net The net benefit of the nomogram predictive scoring system is higher than the net benefit of any one indicator alone. As can be seen in the figure (as Fig 4) below, the net benefit of patients when predicting the surgical outcome of patients based on the nomogram score is shown in red, and it can be seen that the net benefit of patients with the nomogram score is the highest; the net benefit of patients who choose the nomogram score to predict whether or not they will experience a surgical outcome is higher.

**Clinical impact curve analysis (DCA).** In the clinical impact curve(as Fig 5), the red curve indicates the number of people classified as positive (high-risk population) by the

**Table 4. Multiple logistic regression analysis of surgical outcomes.**

| | | Standard Error | OR | 95% confidence interval for OR | | *P* |
|---|---|---|---|---|---|---|
| | | | | Lower limit | Upper limit | |
| Number of surgeries | | 0.191 | 0.563 | 0.387 | 0.819 | 0.003 |
| Mode of surgery | | | | | | <0.001 |
| | Modified Lang method | 1.025 | 16.597 | 2.228 | 123.647 | 0.006 |
| | Inferior triangular flap method | 1.026 | 15.649 | 2.093 | 116.981 | 0.007 |
| | Rotational propulsion method | 1.051 | 3.862 | 0.492 | 30.309 | 0.199 |
| | Other | | 1 | | | |
| Breast milk | Yes | 0.218 | 1.935 | 1.261 | 2.968 | 0.003 |
| | No | | | | | |
| Maternal examination | Done | 0.459 | 2.619 | 1.064 | 6.442 | 0.036 |
| | Not done | | 1 | | | |
| Nutrition during pregnancy | | | | | | 0.017 |
| | Medium | 0.295 | 1.609 | 0.902 | 2.87 | 0.107 |
| | Poor | 0.5 | 3.56 | 1.337 | 9.479 | 0.011 |
| | Good | | 1 | | | |
| Labor intensity during pregnancy | | | | | | 0.003 |
| | Medium | 0.305 | 1.851 | 1.017 | 3.368 | 0.044 |
| | Heavy | 0.245 | 2.152 | 1.331 | 3.481 | 0.002 |
| | Light | | 1 | | | |

SIMPLE model at each threshold probability, and the blue curve shows the number of true positives at each threshold probability, when the threshold probability is low, the difference between the predicted high-risk number and the actual number of positives is large (red and blue distance is large), and as the threshold probability increases, the difference between the predicted high-risk number and the actual number of positives gradually decreases (As the threshold probability increases, the difference between the predicted number of high-risk and the actual number of positives gradually decreases (red-blue distance gradually decreases), and the predicted number of positives and the actual number of positives match when the threshold probability > 40%. The threshold probability used in this study was 40% (i.e., when the probability of a poor surgical outcome was higher than 40%, the patient was predicted to have a poor surgical outcome), meaning that the number of predicted positives and the number of actual positives matched, i.e., using the current nomogram-based predicted poor surgical outcome scoring system, as long as the predicted positives (appearing to have a poor surgical outcome), then the person actually had a greater probability of having a poor surgical outcome.

**Creating a test set to validate accuracy.** The data from 110 patients with external validation data were brought into the score and the diagnostic ROC curve of the nomo score on the surgical outcome was obtained and is shown in Fig 6.

The unsatisfactory and diagnostic value of the nomo score obtained by external validation reached an AUC of 74.5%, p<0.05. This is close to the modeling accuracy of 73.3%, indicating a good validation effect. The predictive accuracy of the predicted versus real surgery was 93.6%.(as Table 7).

## Discussion

Cleft lip and palate is one of the most common congenital defects, which brings a heavy economic burden to patients' families and society. Surgery is the most effective way to treat cleft lip and palate, and as new methods of cleft lip revision are constantly being introduced,

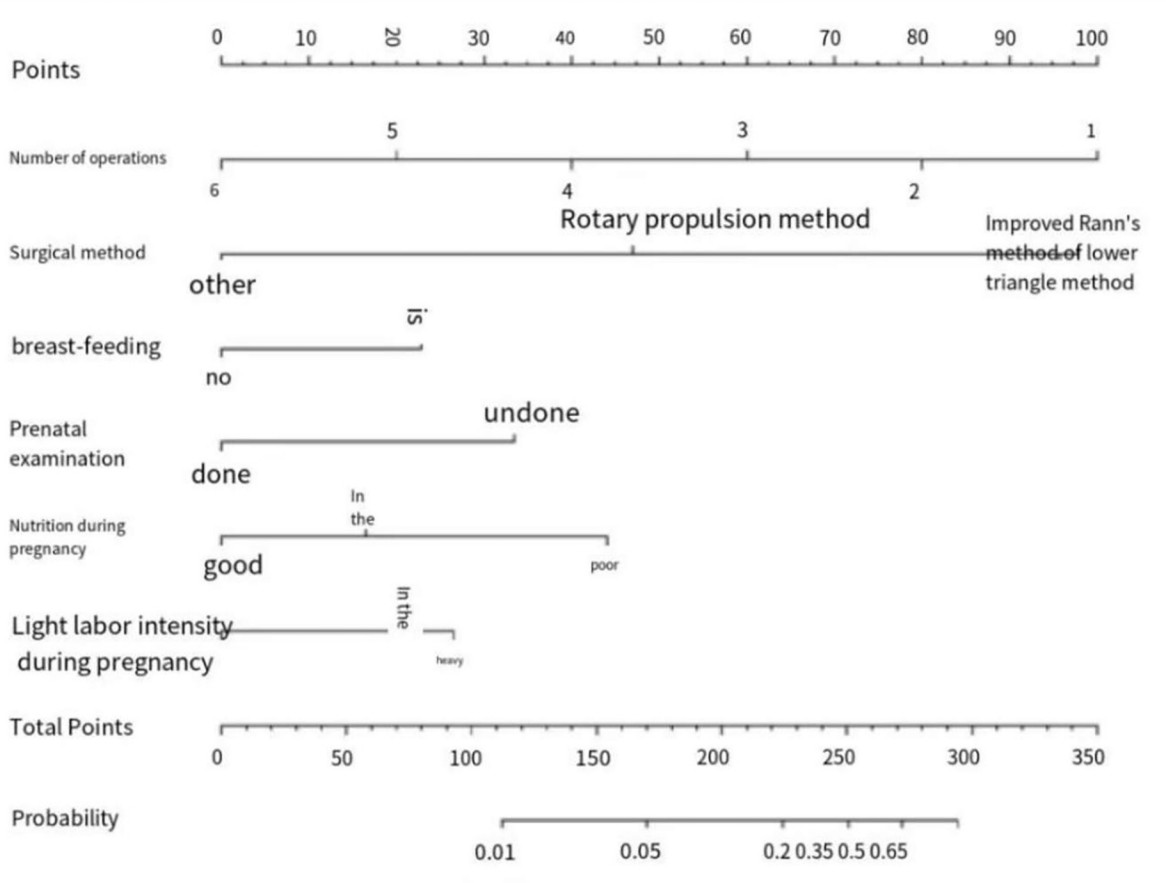

**Fig 1. Surgical outcome prediction score.**

accurate preoperative outcome assessment is of great significance to facilitating preoperative doctor-patient communication, medical strategy development, resource allocation optimization, prognosis judgment, and medical quality improvement. The present study showed that the number of surgeries, surgical method, breast milk, obstetric examination, pregnancy nutrition, and labor intensity during pregnancy were independent risk factors for the poor surgical outcome (all P < 0.05), and based on this analysis using columnar plots and establishing a risk score, the area under the ROC curve (AUC) was 0.733 (95% CI: 0.704 to 0.76) sensitivity. 89.57% Specificity 48.14%. Data from 110 patients with external validation data were brought into the score to establish a predictive scoring system incorporating a total of 5 indicators such as several operations, mode of surgery, breast milk, labor and delivery, nutrition during pregnancy, and labor intensity during pregnancy (all P < 0.05) The critical value was 273 points. The diagnostic value of the under-optimal AUC reached 74.5%, P<0.05, which is close to the modeling accuracy of 73.3%, indicating a good validation effect.

In this study, by screening 26 indicators of cleft lip and palate case data surgically treated in Guiyang Stomatological Hospital, we finally found that the number of surgeries, surgical methods, breast milk, maternal examination, nutrition during pregnancy, and labor intensity during pregnancy were important predictive indicators, and through these important predictive indicators, a predictive evaluation system of cleft lip and palate surgical outcomes was constructed, which can be used for the clinical prediction of surgical outcome of cleft lip and

**Table 5. Surgical outcome prediction score table.**

| Factor | Category | Score |
|---|---|---|
| Number of surgeries | 1 | 100 |
| | 2 | 80 |
| | 3 | 60 |
| | 4 | 40 |
| | 5 | 20 |
| | 6 | 0 |
| Surgical approach | Modified Lang method | 98 |
| | Inferior triangular flap method | 96 |
| | Rotational advancement method | 47 |
| | Other | 0 |
| Breastfeeding | No | 0 |
| | Yes | 23 |
| Antenatal examination | Not Done | 34 |
| | Done | 0 |
| Nutrition during pregnancy | Good | 0 |
| | Medium | 17 |
| | Poor | 44 |
| Labor intensity during pregnancy | Light | 0 |
| | Medium | 21 |
| | Heavy | 27 |

palate patients initially. These predictors reflect the factors affecting surgical outcomes mainly include maternal pregnancy and external reasons for surgery, and this study tried to explore these two factors to improve outcomes of cleft lip and palate surgery and patients' satisfaction. Several indicators from the mother's pregnancy period reflect some extent that the knowledge of the mother of a cleft lip and palate family about eugenics during pregnancy is related to the healthy birth of the child [8], From the perspective of pregnancy nutrition, whether a pregnant woman takes multivitamin supplements and folic acid antagonists in early pregnancy is the

**Table 6. Predicted probability of poor surgical outcome.**

| Overall score | Probability of poor surgical outcome % |
|---|---|
| 113 | 1.00% |
| 170 | 5.00% |
| 196 | 10.00% |
| 212 | 15.00% |
| 224 | 20.00% |
| 234 | 25.00% |
| 243 | 30.00% |
| 251 | 35.00% |
| 259 | 40.00% |
| 266 | 45.00% |
| 273 | 50.00% |
| 280 | 55.00% |
| 287 | 60.00% |
| 294 | 65.00% |
| 302 | 70.00% |

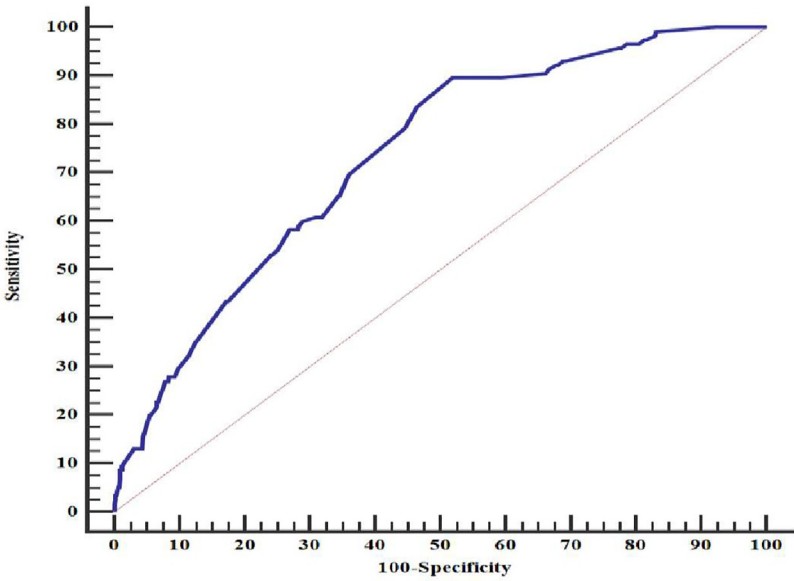

| | AUC | 95% CI | P | sensitivity | 1-Specificity |
|---|---|---|---|---|---|
| NOMO Score | 0.733 | 0.704~0.76 | <0.001 | 89.57% | 48.14% |

**Fig 2. Diagnostic ROC curve of surgical outcome.**

main indicator of nutrition during pregnancy. Studies have shown that folic acid not only protects the fetus from external stimuli during the sensitive period of embryonic development but also plays a significant role in the prevention of neural tube deficiency and CLP in the fetus [9]. From the perspective of labor intensity during pregnancy, most of the subjects in this study come from families in rural areas of Guizhou, and they were engaged in farm work or physical labor during pregnancy until delivery, so we used the labor intensity during pregnancy as an indicator of high-intensity delivery.

From the perspective of breast milk, in the early stages of infant feeding, children with cleft lip and palate have difficulties in breastfeeding compared to normal children because their mouths cannot form a completely closed structure to generate the negative pressure required for effective sucking [10]. The poor feeding situation eventually leads to lower-than-average weight gain, and a high incidence of coughing, vomiting, and respiratory infections, which is related to their nutritional and health status before undergoing surgery and has a strong correlation with the success of the surgery and the prognosis [11]. Therefore, it is more important

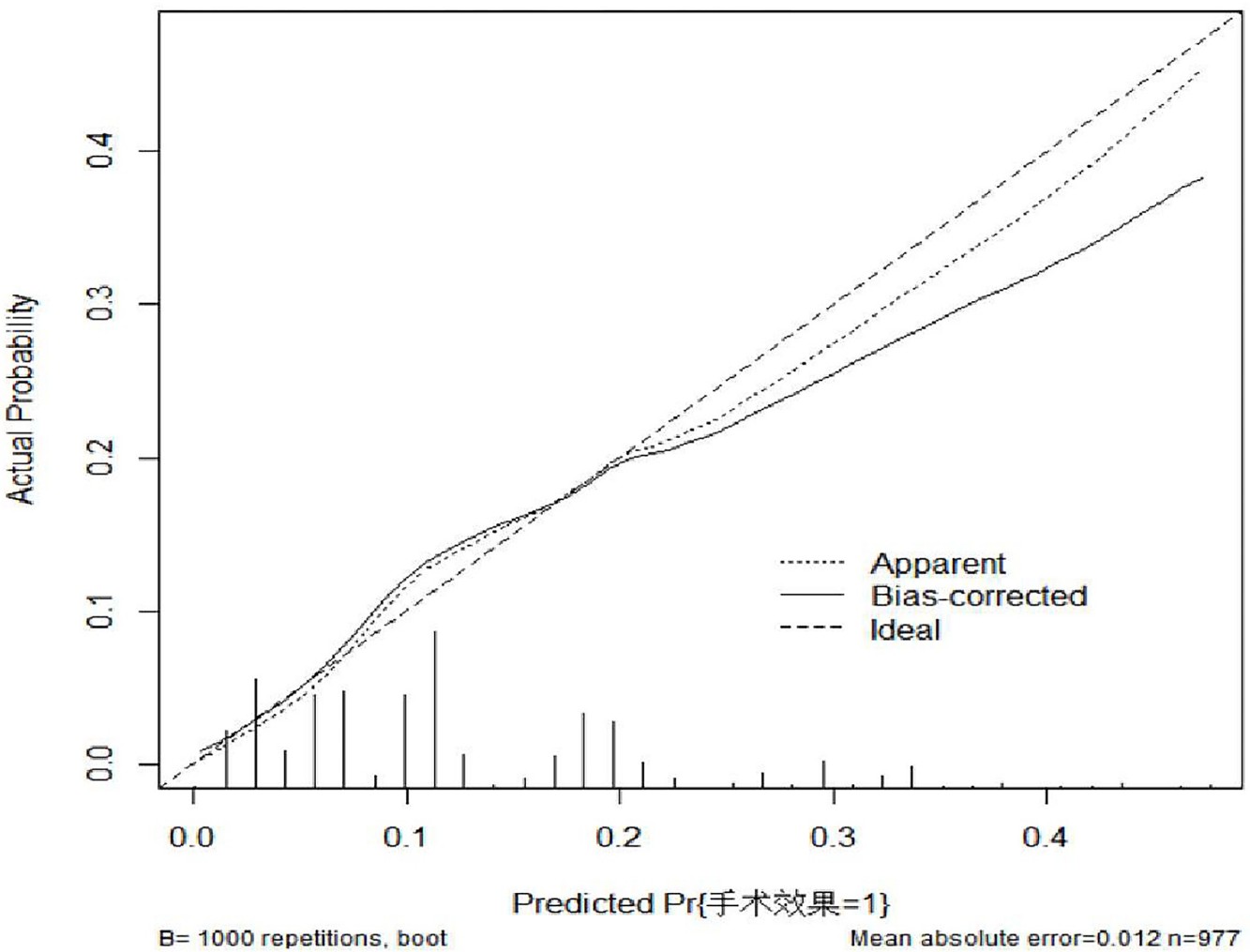

**Fig 3. Scoring system calibration diagram.**

to remind relevant medical staff to start feeding instruction and health education for children with cleft lip and palate as early as possible after birth, focus on instructing parents of children on feeding skills, such as bottle placement and angle, feeding position, and handling of coughing, and start feeding instruction and intervention for children with cleft lip and palate and their parents as early as possible [12] to promote the healthy growth of children with cleft lip and palate.

Some studies have shown that the high incidence of their relatives is related to the proximity of blood relations [13] with significant family aggregation [14],a child with cleft lip and palate is already a heavy burden for these families, and to avoid further cleft lip and palate in the family fetus, the impact of predictive indicators on the surgical outcome can also be more directly enhanced by reminding families of cleft lip and palate patients, especially the mothers, of health education during pregnancy. cleft lip and palate is a congenital malformation of maxillofacial development. It has unique pathological and anatomical characteristics and surgical revision methods, involving the skin, mucosa, muscle, cartilage, and bone of the lip, and most of them need one or several more revisions after the first-stage repair to make their nasolabial close to or reach the normal shape [15]. The degree of deformity of the child, the operating

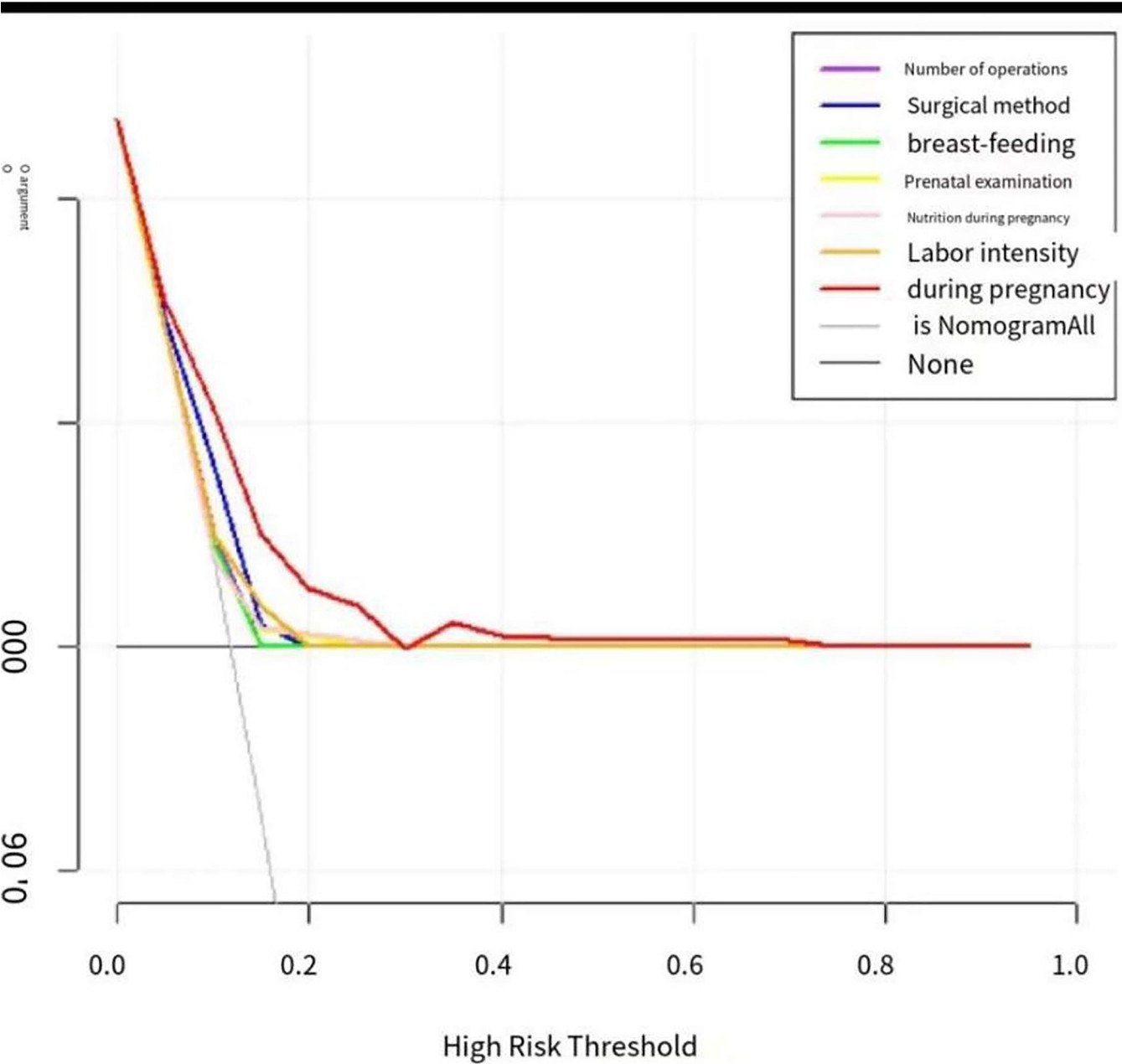

**Fig 4. Decision Curve Analysis (DCA).**

skill of the physician, the cooperation of the anesthesiologist, and the nursing support are necessary for a successful surgery. Therefore, we believe that the clinician should scientifically screen the patient before the surgery and provide the best treatment plan for different patients that are targeted and suitable for individual conditions. Individualized treatment is not a random combination of techniques based on subjective empirical preferences, but follows a standardized design concept, which requires the physician to fully understand the scope of use, advantages, and disadvantages of different technical options. In addition, preoperative communication with the child's family is essential, and the principles of standardized design and

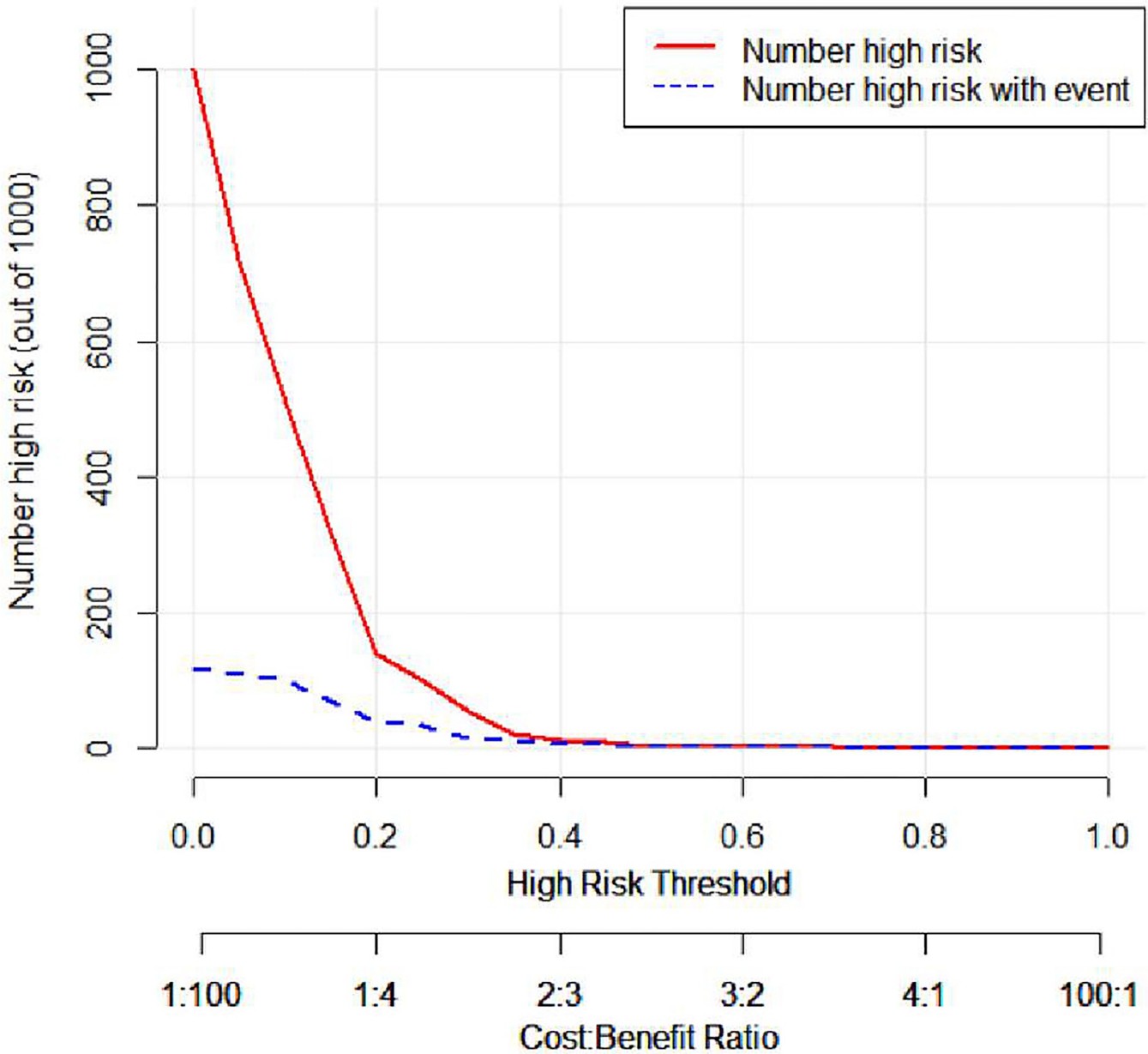

**Fig 5. Clinical impact curve.**

the family's opinions must be taken into account in the selection of surgical methods. In this way, during preoperative doctor-patient communication, the assessment of the preoperative prediction system can be used to help parents establish a good mentality, to improve the quality of doctor-patient communication, to rationalizing the surgical outcomes and helping parents establish a good mindset is beneficial to the recovery and psychological growth of cleft lip and palate patients.

This study shows a preliminary exploration of factors related to the influence of surgical outcome, and there are still some limitations, and the specificity of the included indicators for prediction is still not enough, but this study provides a certain basis for future surgical prediction, and more appropriate indicators can be selected in future studies to establish an accurate,

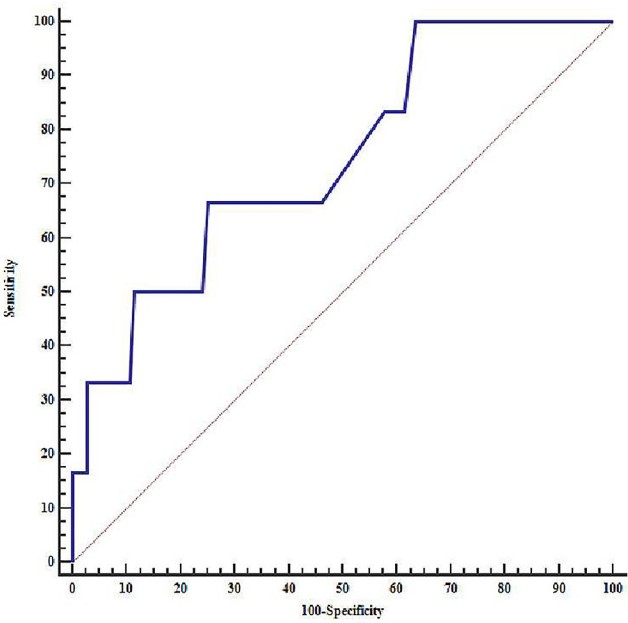

| | AUC | 95%CI | P | sensitivity | 1-Specificity |
|---|---|---|---|---|---|
| NOMO Score | 0.745 | 0.653-0.823 | 0.0253 | 66.67 | 75.00 |

**Fig 6. Test curve.**

reliable, and convenient surgical outcome prediction system to be applied in clinical practice and provide a reference basis for clinicians to carry out cleft lip surgery In the future, we can establish an accurate, reliable and convenient prediction system for clinical practice to provide a reference basis for clinicians to perform cleft lip surgery.Cleft lip revision is a complex systemic project, and continuous.

**Table 7. Comparison of predicted surgical outcome and real surgical outcome.**

| | | Predicted surgical outcome | | Total |
|---|---|---|---|---|
| | | Good | Unsatisfactory | |
| Real surgical outcome | Good | 101 | 3 | 104 |
| | Unsatisfactory | 4 | 2 | 6 |
| Total | | 105 | 5 | 110 |

"Evaluate-improve-reevaluate-improve again" is the only way to continuously improve its revision effect, so that patients can return to society physically and mentally healthy and socially accepted [16], and the surgical prediction index model established in this preliminary study can provide a basis for evaluation and improvement.

## Supporting information

**S1 Data. The dataset used for analyses.**
(XLSX)

## Acknowledgments

We would like to thank all participants in this study.

## Author Contributions

**Conceptualization:** Na Liu, Jingyuan Yang.

**Data curation:** Fang Tan, Haijian Zhu.

**Investigation:** Fang Tan.

**Methodology:** Na Liu.

**Writing – original draft:** Na Liu.

**Writing – review & editing:** Na Liu, Jingyuan Yang.

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
