## [Decision Letter · Decision Letter 0]

20 Mar 2023

PONE-D-23-00885Constructing of Predictive Model for the Surgical Effect of Patients with Cleft Lip and PalatePLOS ONE

Dear Dr. Yang,

Thank you for submitting your manuscript to PLOS ONE. After careful consideration, we feel that it has merit but does not fully meet PLOS ONE’s publication criteria as it currently stands. Therefore, we invite you to submit a revised version of the manuscript that addresses the points raised during the review process.

I would like to sincerely apologise for the delay you have incurred with your submission. It has been exceptionally difficult to secure reviewers to evaluate your study. We have now received two completed reviews; the comments are available below. The reviewers have raised significant scientific concerns about the study that need to be addressed in a revision.

Please revise the manuscript to address all the reviewer's comments in a point-by-point response in order to ensure it is meeting the journal's publication criteria. Please note that the revised manuscript will need to undergo further review, we thus cannot at this point anticipate the outcome of the evaluation process.

We look forward to receiving your revised manuscript.

Kind regards,

Miquel Vall-llosera Camps

Senior Editor

PLOS ONE

“The authors have declared that nocompeting interests exist.

The funders had no role in study design,data collection and analysis,decisionto publish, or preparation of the manuscript.”

3. Please amend the manuscript submission data (via Edit Submission) to include authors Fang Tan and   Haijian Zhu.

5. Please include your tables as part of your main manuscript and remove the individual files. Please note that supplementary tables (should remain/ be uploaded) as separate "supporting information" files

Reviewers' comments:

Reviewer's Responses to Questions

**Comments to the Author**

1. Is the manuscript technically sound, and do the data support the conclusions?

Reviewer #1: No

Reviewer #2: No

2. Has the statistical analysis been performed appropriately and rigorously? 

Reviewer #1: Yes

Reviewer #2: I Don't Know

3. Have the authors made all data underlying the findings in their manuscript fully available?

Reviewer #1: No

Reviewer #2: Yes

4. Is the manuscript presented in an intelligible fashion and written in standard English?

Reviewer #1: Yes

Reviewer #2: No

5. Review Comments to the Author

Reviewer #1: Dear authors.

I like to provide you some observations regarding your interesting study.

An important limitation are the groups conformation and follow up time.

Outcomes have been considered as good or poor however these were estimated using subjective scales like Asher Mc Dade. In addition, there is not a correlation between the categories of used scales and your categories (good and poor).

It is not clear how are you considering the outcomes as good and poor in this study.

In addition, the follow up time used in this study is not adequate to evaluate a surgical outcome, this should be at least one year.

Finally, the most important predictive factor is probably the surgeon performance and this was not included. In addition, should be important to analyze the clinical relevance of the statistical significant factors like nutritional ones.

Reviewer #2: Thank you for providing me with the opportunity to review the manuscript entitled “Constructing of Predictive Model for the Surgical Effect of Patients with Cleft Lip and Palate” for consideration for publication in PLOS One.

Here, the authors have created a predictive model to evaluate post-operative surgical aesthetic outcomes in cleft lip and nasal deformity repair in a rather large cohort of patients between 2015 – 2020. The number of patients and data incorporated into the model is impressive and the predictive nature of it impressive as well; however, it does raise many questions and comments.

1. The article would benefit from significant revision to enhance readability.

2. My largest concern is related to the model.

a. There is a significant amount of data included in the model with many confounders that have not been addressed.

b. How did the authors decide on what was an intense labor? What about nutrition? Did the authors have data on intensity of labor in a 40 year old? Im assuming the data was incomplete for some of these patients.

c. Were cleft lips (incomplete) compared against complete cleft lips? LAHSHAL nomenclature or some standardized nomenclature would be needed. One cannot compare an incomplete cleft lip with a spared alveolus to a complete cleft lip and palate. Was NAM available or some form of presurgical molding? Was syndromic status taken into consideration?

d. The model seems to include revisions as well. This raises another question about homogeneity when it comes to the model. The model includes patients of all ages, cleft types, and revision status. This introduces a significant amount of variability as it ignores the importance of facial growth. Number of surgeries would certainly correlate with a poor result – the fact that one is operating more than once on a lip means the index procedure may not have been as ideal. Issues such as these do not seem to have been taken into considering in the statistical model.

e. Cleft lip and palate outcomes assessment is complex. There are many rating scales and there is a lot of room for subjectivity. While the authors have used somewhat standardized scales, there is still room for significant subjectivity.

f. The authors discuss using this in the preoperative consultation. What actions can a provider take to improve the outcome? Based on the data, surgical method and breast feeding are the only two “modifiable” factors that can be changed after a child is born. Do the authors recommend that they care giver abstain from breast-feeding (I would not recommend this based on the wealth of information supporting the significant benefits associated with breast feeding, especially in resource-constrained settings? How this is used in the clinic is not explored. I would be worried that, as constructed, this could become somewhat of a self-fulfilling prophecy.

g. Surgical method was important. What method should be used? This is not really discussed.

Tables and figures should be referenced in the manuscript in order. For example, Table 3 comes before Table 1.

Thank you very much for this opportunity.

6. PLOS authors have the option to publish the peer review history of their article (what does this mean?). If published, this will include your full peer review and any attached files.

Reviewer #1: **Yes: **Percy ROSSELL-PERRY

Reviewer #2: No

---

## [Author Response · Author response to Decision Letter 0]

23 May 2023

RE: 

Constructing of Predictive Model for the Surgical Effect of Patients with Cleft Lip and Palate

(PONE-D-23-00885)

Dear Editors and Reviewers,

Thanks a lot for giving us the opportunity to revise our manuscript. We express our sincere thanks to the reviewers for the constructive and thoughtful comments on previous draft. Those comments are all valuable and very helpful for revising and improving our paper. We have studied comments carefully and have made correction. The revised manuscript is highlighted with a yellow background in the revised manuscript. 

We hope the revision is acceptable and we look forward to hearing from you soon.

Once again thank you very much for your comments and suggestions.

With best wishes.

Yours sincerely,

 Jingyuan Yang

Reviewer #1 (Comments to the Authors)

An important limitation are the groups conformation and follow up time.

Response: Thanks to the referee for the good review and kind suggestion.

The evaluation team was mainly composed of a physician with a senior title, a physician with an intermediate or junior title, and a nursing staff, and the tracking lasted for one year through outpatient follow-up, remote video, or photo follow-up. We did it mainly because most of the cleft lip and palate patients come from poor families in remote rural mountainous areas of Guizhou, and their surgical costs and travel expenses come from the national major disease program for cleft lip and palate patients. Therefore, the travel expenses for postoperative follow-up are an additional financial burden for the patient’s families. We followed up with patients to the maximum extent possible to save them from traveling. We finished one outpatient follow-up and two remote video and photo follow-ups within one year after surgery.

Add the above changes to“Structure of the surgical outcome evaluation team and tracking time”on Page4，Lines 81-91.

Reviewer #1 (Comments to the Authors)

Outcomes have been considered as good or poor however these were estimated using subjective scales like Asher Mc Dade. In addition, there is not a correlation between the categories of used scales and your categories (good and poor).

Response: Thanks to the referee for the good review and kind suggestion.

As stated by the experts, our evaluation has a certain degree of subjectivity, which is an objective problem for us, and we will also incorporate more rigorous evaluation indicators in our later studies, but for this study, we mainly followed the following indicators for the reference of surgical outcomes: 1. Measurements were taken through preoperative and postoperative photos to determine whether the symmetry and aesthetics of the nasolabial shape were restored after cleft lip surgery; the degree of velopharyngeal closure, the function of an oropharyngeal area and the verbal ability of cleft palate patients; 2. The satisfaction of patients and their families with the surgery. The evaluation team scored based on the above indicators and finally made a comprehensive determination of the evaluation results of the three. Therefore, the judgment of this study still has a certain scientific basis.

Reviewer #1 (Comments to the Authors)

In addition, the follow up time used in this study is not adequate to evaluate a surgical outcome, this should be at least one year.

Response: Thanks to the referee for the good review and kind suggestion. 

The tracking in this study lasted for one year through outpatient follow-up, remote video, or photo follow-up. We did it mainly because most of the cleft lip and palate patients come from poor families in remote rural mountainous areas of Guizhou, and their surgical costs and travel expenses come from the national major disease program for cleft lip and palate patients. Therefore, the travel expenses for postoperative follow-up are an additional financial burden for the patient’s families. We followed up with patients to the maximum extent possible to save them from traveling. We finished one outpatient follow-up and two remote video and photo follow-ups within one year after surgery.

Reviewer #1 (Comments to the Authors)

Finally, the most important predictive factor is probably the surgeon performance and this was not included. In addition, should be important to analyze the clinical relevance of the statistical significant factors like nutritional ones.

Response: Thanks to the referee for the good review and kind suggestion. 

In this study, the surgeon's performance was attributed to the surgical method as a predictor. Due to data on surgeon performance intraoperatively could not be collected objectively, we believe that physicians' preoperative scientific provision of targeted, optimal treatment plans and surgical methods for different patients that are appropriate for individual conditions is an important factor in surgical outcomes, but it was generally finished according to existing norms and was not included in this analysis because differences involving individual protocols are not comparable.

In the "Discussion" on Page19,Lines263-267, we revised according to your advice as follows: the degree of deformity of a child, the operating skill of the physician, the cooperation of the anesthesiologist, and the nursing support are necessary for a successful surgery. Therefore, we believe that the clinician should scientifically screen the patient before the surgery and provide the best treatment plan for different patients that are targeted and suitable for individual conditions.

As for nutritional factors, we focused on statistical analysis from the correlation of nutrition during pregnancy and conducted a comprehensive evaluation of whether a pregnant woman takes multivitamin supplements and folic acid antagonists in early pregnancy as the main indicator.

In the " Discussion" on Page18,Lines 234-238, we revised according to your advice as follows:

From the perspective of pregnancy nutrition, whether a pregnant woman takes multivitamin supplements and folic acid antagonists in early pregnancy is the main indicator of nutrition during pregnancy. Studies have shown that folic acid not only protects the fetus from external stimuli during the sensitive period of embryonic development but also plays a significant role in the prevention of neural tube deficiency and CLP in the fetus.

Reviewer #2 (Comments to the Authors)

1. The article would benefit from significant revision to enhance readability.

Response: Thanks to the referee for the good review and kind suggestion. 

 The article details have been revised according to this revised version.

Reviewer #2 (Comments to the Authors)

2.a. There is a significant amount of data included in the model with many confounders that have not been addressed.

Response: Thanks to the referee for the good review and kind suggestion. 

We also realized the problem you mentioned. This study conducted statistical analysis from a large sample perspective and used a multi-factor model analysis to control potential confounding factors to some extent, but the variables included in the model were still relatively limited. Because most of our cleft lip and palate patients come from poor families in remote rural mountainous areas of Guizhou, this construction model still provides a scientific basis to convince more children with cleft lip and palate to receive surgical treatment as early as possible.

b. How did the authors decide on what was an intense labor? What about nutrition? Did the authors have data on intensity of labor in a 40 year old? Im assuming the data was incomplete for some of these patients.

Response: Thanks to the referee for the good review and kind suggestion. 

Most of the subjects in this study (cleft lip and palate patients) were from families in rural areas of Guizhou, and they were engaged in farm work or physical labor during pregnancy until delivery, so we used the labor intensity during pregnancy as an indicator of high-intensity delivery, with mild being those who worked in a sitting position, moderate being those who bore load <20 kg each time and worked in a standing position, and severe being those who bore load >20 kg each time, and these data were obtained mainly through questionnaires (interviews) with the mothers.

In the "Discussion" on Page18 Lines238-242, we modified it according to your advice as follows:

From the perspective of labor intensity during pregnancy, most of the subjects in this study come from families in rural areas of Guizhou, and they were engaged in farm work or physical labor during pregnancy until delivery, so we used the labor intensity during pregnancy as an indicator of high-intensity delivery.

As for nutritional factors, we focused on statistical analysis from the correlation of nutrition during pregnancy and conducted a comprehensive evaluation of whether a pregnant woman takes multivitamin supplements and folic acid antagonists in early pregnancy as the main indicator.

In the "Discussion" Page18,Lines 234-238, we revised according to your advice as follows: From the perspective of pregnancy nutrition, whether a pregnant woman takes multivitamin supplements and folic acid antagonists in early pregnancy is the main indicator of nutrition during pregnancy. Studies have shown that folic acid not only protects the fetus from external stimuli during the sensitive period of embryonic development but also plays a significant role in the prevention of neural tube deficiency and CLP in the fetus.

c. Were cleft lips (incomplete) compared against complete cleft lips? LAHSHAL nomenclature or some standardized nomenclature would be needed. One cannot compare an incomplete cleft lip with a spared alveolus to a complete cleft lip and palate. Was NAM available or some form of presurgical molding? Was syndromic status taken into consideration?

Response: Thanks to the referee for the good review and kind suggestion. 

In this study, due to the large sample size and many variables, our comparison of patients' surgical outcomes was mainly on the overall situation after individual surgery. Measurements were taken through preoperative and postoperative photos to determine whether the symmetry and aesthetics of the nasolabial shape were restored after cleft lip surgery, the degree of velopharyngeal closure, the function of an oropharyngeal area, and the verbal ability of cleft palate patients, as well as the degree of satisfaction of patients and their families with the surgery. This study focused on several factors that may have an impact on the outcome of cleft lip and cleft palate surgery. Therefore, no specific comparison of surgical outcomes for the diagnosis of cleft lip or cleft palate was performed. We considered syndromic status, and all cases were non-syndromic cleft lip and palate patients, as mentioned in the "Materials and methods : non-syndromic cleft lip and palate patients".( Page3,Lines63-64)

d. The model seems to include revisions as well. This raises another question about homogeneity when it comes to the model. The model includes patients of all ages, cleft types, and revision status. This introduces a significant amount of variability as it ignores the importance of facial growth. Number of surgeries would certainly correlate with a poor result – the fact that one is operating more than once on a lip means the index procedure may not have been as ideal. Issues such as these do not seem to have been taken into considering in the statistical model.

Response: Thanks to the referee for the good review and kind suggestion. 

There are still problems in the data analysis as stated by the experts. There is variability in facial growth, and the number of surgeries introduces a corresponding bias. The proportion of subjects with multiple surgeries in this study was small and was excluded at the time of the study. The variability of facial growth may be a long-term process that is difficult to assess, and this follow-up was within 1 year, so the effect of facial growth can be considered minor, and the advice of experts has important reference value for future studies by our group.

e. Cleft lip and palate outcomes assessment is complex. There are many rating scales and there is a lot of room for subjectivity. While the authors have used somewhat standardized scales, there is still room for significant subjectivity.

Response: Thanks to the referee for the good review and kind suggestion. 

Finally, our evaluation has a certain degree of subjectivity, which is an objective problem for us, and we will also incorporate more rigorous evaluation indicators in our later studies, but for this study, we mainly followed the following indicators for the reference of surgical outcomes: 1. Measurements were taken through preoperative and postoperative photos to determine whether the symmetry and aesthetics of the nasolabial shape were restored after cleft lip surgery; the degree of velopharyngeal closure, the function of an oropharyngeal area and the verbal ability of cleft palate patients; 2. The satisfaction of patients and their families with the surgery. The evaluation team scored based on the above indicators and finally made a comprehensive determination of the evaluation results of the three. 

We have added the above changes to " Structure of the surgical outcome evaluation team and tracking time."(Page4,Lines81-91)

f. The authors discuss using this in the preoperative consultation. What actions can a provider take to improve the outcome? Based on the data, surgical method and breast feeding are the only two “modifiable” factors that can be changed after a child is born. Do the authors recommend that they care giver abstain from breast-feeding (I would not recommend this based on the wealth of information supporting the significant benefits associated with breast feeding, especially in resource-constrained settings? How this is used in the clinic is not explored. I would be worried that, as constructed, this could become somewhat of a self-fulfilling prophecy.

Response: Thanks to the referee for the good review and kind suggestion. 

First, we absolutely support breastfeeding and strongly advocate and agree with breastfeeding. However, the rate of poor surgical outcome of breastfed children with cleft lip and palate in the study was 1.935 times higher than that of non-breastfed, which is clearly contrary to a large number of objective studies, and our analysis suggests that it may be due to the following reasons: First, our data collection on breastfeeding was mainly before the surgery, in the early stages of infant feeding, children with cleft lip and palate have difficulties in breastfeeding compared to normal children because their mouths cannot form a completely closed structure to generate the negative pressure required for effective sucking. The poor feeding situation eventually leads to lower-than-average weight gain, and a high incidence of coughing, vomiting, and respiratory infections, which is related to their nutritional and health status before undergoing surgery and has a strong correlation with the success of the surgery and the prognosis. Second, we believe that the concluding data should not be generalized to breast milk and non-breast milk. Therefore, it is more important to remind relevant medical staff to start feeding instruction and health education for children with cleft lip and palate as early as possible after birth, focus on instructing parents of children on feeding skills, such as bottle placement and angle, feeding position, and handling of coughing, and start feeding instruction and intervention for children with cleft lip and palate and their parents as early as possible to promote the healthy growth of children with cleft lip and palate.

We have added the above changes to “Discussion" on Page18,Lines243-254.

g. Surgical method was important. What method should be used? This is not really discussed.

Response: Thanks to the referee for the good review and kind suggestion. 

The degree of deformity of a child, the operating skill of the physician, the cooperation of the anesthesiologist, and the nursing support are necessary for a successful surgery. Therefore, we believe that the clinician should scientifically screen the patient before the surgery and provide the best treatment plan for different patients that are targeted and suitable for individual conditions. Individualized treatment is not a random combination of techniques based on subjective empirical preferences, but follows a standardized design concept, which requires the physician to fully understand the scope of use, advantages, and disadvantages of different technical options. In addition, preoperative communication with the child's family is essential, and the principles of standardized design and the family's opinions must be taken into account in the selection of surgical methods.

We have added the above changes to "Discussion" on Page19,Lines263-272.

Tables and figures should be referenced in the manuscript in order. For example, Table 3 comes before Table 1.

Response: Thanks to the referee for the good review and kind suggestion. 

 Adjustments have been made accordingly.

---

## [Editor Report · Decision Letter 1]

29 May 2023

Constructing of Predictive Model for the Surgical Effect of Patients with Cleft Lip and Palate

PONE-D-23-00885R1

Dear Dr. Yang,

We’re pleased to inform you that your manuscript has been judged scientifically suitable for publication and will be formally accepted for publication once it meets all outstanding technical requirements.

Kind regards,

Engy Asem Ashaat

Academic Editor

PLOS ONE
---

## [Editor Report · Acceptance letter]

22 Jun 2023

PONE-D-23-00885R1 

Constructing of Predictive Model for the Surgical Effect of Patients with Cleft Lip and Palate 

Dear Dr. Yang:

I'm pleased to inform you that your manuscript has been deemed suitable for publication in PLOS ONE. Congratulations! Your manuscript is now with our production department. 

Kind regards, 

on behalf of

Professor Engy Asem Ashaat 

Academic Editor

PLOS ONE